# HRAMS Proteomics Insights on the Anti-Filarial Effect of *Ocimum sanctum*: Implications in Phytochemical-Based Drug-Targeting and Designing

**DOI:** 10.3390/proteomes13010002

**Published:** 2024-12-27

**Authors:** Ayushi Mishra, Vipin Kumar, Sunil Kumar, HariOm Singh, Anchal Singh

**Affiliations:** 1Department of Biochemistry, Institute of Science, Banaras Hindu University, Varanasi 221005, India; ayushimishr14@gmail.com (A.M.);; 2Department of Molecular Biology, National Aids Research Institute, Pune 411026, India

**Keywords:** bioactive compounds, lymphatic filariasis treatment, anti-filarial compounds, rutin, lead anti-filarial, anti-filarial drug targeting

## Abstract

Lymphatic filariasis (LF) continues to impact 657 million individuals worldwide, resulting in lifelong and chronic impairment. The prevalent anti-filarial medications—DEC, albendazole, and ivermectin—exhibit limited adulticidal efficacy. Despite ongoing LF eradication programs, novel therapeutic strategies are essential for effective control. This study examines the mechanism of action of *Ocimum sanctum* on the filarial parasites *Setaria cervi* via a synergistic biochemical and proteomics methodology. The ethanolic extract of *Ocimum sanctum* (EOS) demonstrated potential anti-filarial action in the MTT reduction experiment, with an LC_50_ value of 197.24 µg/mL. After EOS treatment, an elevation in lipid peroxidation (51.92%), protein carbonylation (48.99%), and NADPH oxidase (88.88%) activity, along with a reduction in glutathione (GSH) (−39.23%), glutathione reductase (GR) (−60.17%), and glutathione S transferase (GST) (−50.48%) activity, was observed. The 2D gel electrophoresis identified 20 decreased and 11 increased protein spots in the EOS-treated parasites relative to the control group. Additionally, in drug docking analysis, the EOS bioactive substances ursolic acid, rutin, and rosmarinic acid show a significant binding affinity with the principal differentially expressed proteins. This paper demonstrates, for the first time, that the anti-filarial efficacy of EOS is primarily facilitated by its impact on energy metabolism, antioxidant mechanisms, and stress response systems of the parasites.

## 1. Introduction

Lymphatic filariasis (LF) is a neglected tropical disease referred to as elephantiasis. At present, 657 million individuals are at risk of lymphatic filariasis infection throughout 39 endemic countries [1]. The lymphatic-dwelling parasite roundworms, namely, *Wuchereria bancrofti*, *Brugia malayi*, and *Brugia timori*, are the etiological agents of the disease. LF is responsible for severe morbidity, especially in the tropical and sub-tropical regions of the world. The disease accounts for an estimated 5.5 million disability adjusted life years (DALY) globally [2]. LF leads to chronic and long-term disability and poses a substantial obstacle to the socioeconomic advancement of affected individuals in endemic areas. In 2020, the WHO reset the deadline of the Global Programme to Eliminate Lymphatic Filariasis (GPELF) from 2020 to 2030 by stopping the spread of infection. The WHO recommends preventive chemotherapy via mass drug administration (MDA) of anti-filarial medicines to the community at risk of LF infection. The MDA program includes annual doses of the anti-filarial drugs diethylcarbamazine (DEC), ivermectin, and albendazole, either in combination or alone. However, these drugs have only microfilaricidal activity and are unable to kill the adult worms, which typically reside inside the host lymphatics [3]. In addition to the absence of microfilaricidal action, anti-filarial medications are linked to several side effects. DEC and ivermectin can cause adverse effects in microfilaremic patients, i.e., fever, headache, myalgia, fatigue, and malaise. Additionally, DEC and ivermectin are not recommended when LF infection is co-endemic with onchocerciasis or loiasis, as loss of vision and fatal encephalopathy has been reported in certain instances [4]. Administration of albendazole is associated with vomiting, headache, temporary hair loss, nausea, fever, and dizziness [5], which deter patients from subsequent doses. Compounding these adversities is the emergence of anti-helminthic drug resistance in parasitic nematodes, which has now become a major concern for LF treatment [6]. Consequently, there is an urgent necessity for the formulation of effective macrofilaricidal agents that are non-toxic and exhibit minimum adverse effects.

Plants serve as a rich source of active components and secondary metabolites; moreover, phytomedicines against many chronic and parasitic diseases are already in use or under clinical trial [7]. Recently, more traditional medicinal plants and their active ingredients are being explored for development of new drugs for chronic infections and illnesses due to their reduced toxicity and adverse effects. *Ocimum sanctum* (OS), an aromatic plant belonging to the *Lamiaceae* family, is prevalent in tropical and subtropical regions of Asia, including India, and is extensively utilized in traditional medicines. OS has antioxidant, antimicrobial, anti-inflammatory, and immunomodulatory properties [8,9]. The major OS bioactive compounds are eugenol, rosmarinic acid, ursolic acid, and kaempferol, which contribute to its pharmacological effects [10,11].

In a previous study, the anti-filarial effect of OS was evaluated on lymphatic filarial parasites *Setaria cervi* [12]. The present study was undertaken to investigate the molecular mechanism of the anti-filarial effect of OS using a combined proteomics, biochemical, and in silico approach. Although *S. cervi* is a bovine filarial parasite, its opportunistic and accidental infections in humans leads to neurological syndromes, lung infections, abscess, and allergic reactions [13]. In the present study, the effect of ethanolic extract of *Ocimum sanctum* (EOS) was evaluated on parasite viability, augmentation of ROS parameters, antioxidant enzymes/proteins like GST, GR, GSH, and NADPH oxidase. Further, 2D electrophoresis and high-resolution accurate mass spectrometry (HRAMS) analysis were used to evaluate the impact of EOS on the *S. cervi* proteoforms profile. This research is the first study elucidating the molecular mechanism underlying the anti-filarial effects of *Ocimum sanctum*, which may be further investigated for the formulation of novel anti-filarial treatments.

## 2. Materials and Methods

### 2.1. Ethanolic Extract Preparation of Ocimum sanctum

Ethanolic extract of Ocimum sanctum (EOS) was prepared according to Mishra et al. [12]. The leaves were washed, shade dried, prepared to a fine powder with a mortar and pestle, and stored at 4 °C for extract extraction. A 10% ethanol extract was made from OS leaf powder using a Soxhlet device. The ethanol was evaporated with a rotary evaporator, and the dried extract was kept at −20 °C. The dried OS leaf extract was weighed and solubilized in DMSO (100 mg/mL concentration) before each application.

### 2.2. Parasite Collection and Culture

*S. cervi* nematodes were obtained from the peritoneal folds of Indian water buffaloes slaughtered for non-vegetarian cuisines. Adult live female *S. cervi* were transported to the laboratory in Krebs’s Ringer bicarbonate buffer (KRB), containing 100 U/mL streptomycin, 100 µg/mL penicillin, 2 mM glutamine, and 0.5% glucose (maintenance medium). Worms were rinsed with KRB and incubated in a maintenance medium at 37 °C in a water bath for one hour prior to any subsequent treatments [14]. The *S. cervi* worms (n = 20) were subjected to varying concentrations of EOS, i.e., 125 μg/mL, 250 μg/mL, and 375 μg/mL, for a duration of 6 h at 37 °C in a CO_2_ incubator in triplicates. Worms incubated in maintenance medium containing DMSO (0.37%) acted as the vehicle control.

### 2.3. Preparation of S. cervi Homogenate

A total of 10% *w*/*v* homogenate of adult female *S. cervi* was prepared in 100 mM Tris-HCl, pH 7.0, including 1 mM EDTA and 1 mM phenylmethylsulphonyl fluoride (PMSF), utilizing a motor-driven homogenizer (REMI type RQ127A, Mumbai, India) at 4 °C [15]. The control and EOS-treated parasites were separately homogenized in 100 mM Tris–HCl, pH 7.0, with 1 mM PMSF and 1 mM EDTA, using a REMI homogenizer model RQ 127A at 4 °C, followed by centrifugation at 5000× *g* and subsequently at 12,000× *g* for 30 min. Further, centrifugation at 1000× *g* for 10 min at 4 °C was performed. The resultant clear supernatant was preserved at −20 °C in aliquots after the addition of protease inhibitors PMSF and 1 mM EDTA. Protein quantification was conducted using Bradford’s method [16], with bovine serum albumin (BSA) serving as the reference.

### 2.4. Estimation of Viability

The viability of control and EOS-treated *S. cervi* parasites was assessed using the MTT assay [17] in three independent sets. *S. cervi* worms (n = 5) were cultured in PBS medium with 0.5 mg/mL MTT (3-(4, 5-dimethylthiazol-2-yl)-2, 5-diphenyl tetrazolium bromide) for 2 h at 37 °C in darkness. Subsequently, the worms were introduced into 200 µL of dimethyl sulfoxide (DMSO) to solubilize the formazan crystals. After 1 h, the OD of the solution was assessed at 540 nm using a microplate reader (BioRad, Hercules, CA, USA).

### 2.5. Estimation of Glutathione (GSH) Level

The GSH level was assessed thrice in *S. cervi* worm extract using Ellman’s technique [18], with few changes. Equal amounts of *S. cervi* extract and 5% meta phosphoric acid (MPA) were combined and centrifuged at 1000× *g* at 4 °C for 10 min. The supernatant was subsequently isolated, and 100 µL of it was combined with 1.88 mL of 0.1 M potassium phosphate buffer (pH 8.0) and 0.02 mL of 4% DTNB (5,5′-dithiobis-(2-nitrobenzoic acid)), resulting in a final volume of 2 mL. The reaction mixture was incubated for 15 min at ambient temperature, and the absorbance was spectrophotometrically measured at 412 nm. GSH concentrations were determined using a standard curve of reduced glutathione. The experiment was run in triplicates and repeated thrice.

### 2.6. Estimation of Glutathione-S-Transferase (GST) Activity

The GST activity was assessed using the method of Habig et al. [19] in both control and treated *S. cervi* worms. In summary, 50 µL of extract was combined with 0.1 M phosphate buffer (pH 6.5), 2 mM GSH, and 1 mM CDNB (1-chloro-2,4-dinitrobenzene) to achieve a final volume of 1 mL at 25 °C. The absorbance was recorded for 3 min at 340 nm. One unit of GST enzyme activity is defined as the quantity of enzyme that catalyzes the oxidation of 1 mM of substrate per mL per minute at 25 °C. The experiment was run in triplicate and repeated thrice.

### 2.7. Estimation of Glutathione Reductase (GR) Activity

The GR activity was evaluated by combining 0.1 mM NADPH with 50 µL of *S. cervi* extract in a 50 mM potassium phosphate buffer (pH 7.0), 0.2 mM EDTA, and 0.5 mM oxidized glutathione (GSSG) at ambient temperature. The variation in absorbance was observed with a Systronics UV-Vis spectrophotometer for 3 min at 340 nm. One unit of GR enzyme activity is defined as the quantity of enzyme that catalyzes the reduction of 1 µmol of NADPH per minute (ε 340 nm for NADPH is 6.22 mM^−1^ cm^−1^) [20]. Independent experiments were performed in three sets and repeated thrice.

### 2.8. Estimation of Thioredoxin Reductase (TrxR) Activity

The activity of the TrxR enzyme in the EOS-treated extract was assessed in triplicates in three independent experiments via the reduction of DTNB in the presence of NADPH [21]. The assay mixture comprised 0.1 M potassium phosphate buffer (pH 7.0) and 2 mM EDTA, in a final volume of 1 mL, combined with 10 μM NADPH and 50 μM DTNB. The reaction commenced with the addition of NADPH, and the absorbance increase at 412 nm was recorded for 3 min at 25 °C. One unit of enzyme activity is defined as the NADPH-dependent synthesis of 2 μmol of 2-nitro-5-thiobenzoate (ε412 nm 13.6 mM^−1^ cm^−1^) per min.

### 2.9. Estimation of Protein Carbonyl Content

The parasites were monitored by assessing the protein carbonyl concentration in three independent sets [22]. A total of 10% cold trichloroacetic acid (TCA) solution was combined with worm extract in a 1:1 ratio and centrifuged at 6000× *g* for 5 min at 4 °C. Subsequently, the pellet was reconstituted in 10 mM DNPH (2,4-Dinitrophenylhydrazine) formulated in 2 N HCl and incubated at ambient temperature for 1 h with intermittent vortexing. Centrifugation was conducted at 6000× *g* for 5 min at 4 °C, after which 20% TCA was added to the pellet, followed by a second centrifugation at 6000× *g* for 5 min at 4 °C. The pellet was subsequently washed three times with a combination of ethanol and ethyl acetate (1:1), after which it was resuspended in 800 µL of 6 M guanidium hydrochloride, and absorbance was measured at 370 nm. The molar extinction coefficient of 22,000 × 106 μM^−1^ cm^−1^ was utilized for computations.

### 2.10. Estimation of Lipid Peroxidation

Lipid peroxidation was assessed in three replicates by quantifying malondialdehyde (MDA) levels [23]. In summary, 100 µL of 10% SDS was combined with 300 µL of both control and treated *S. cervi* extract, followed by a 5 min incubation at room temperature. Subsequently, 600 µL of 20% acetic acid was incorporated into the reaction mixture, and after 2 min, 600 µL of 0.8% 2-thiobarbituric acid (TBA) was pipetted, resulting in a final volume of 3 mL. The reaction mixture was subsequently heated for 1 h in a water bath and then cooled to 4 °C. Next, it was centrifuged at 10,000× *g* for 5 min at 4 °C, and the absorbance of the supernatant containing active TBA reactive substances was measured at 532 nm. The molar extinction coefficient of 1.53 × 105 M^−1^ cm^−1^ for MDA was utilized in the computations.

### 2.11. Estimation of NADPH Oxidase Activity

The NADPH oxidase activity experiment was conducted in three sets by homogenizing five worms in a 50 mM phosphate buffer (pH 7.2) with 0.25% SDS. The mixture underwent centrifugation at 600× *g* for 10 min at 4 °C. Additionally, 100 µL of the supernatant was combined with 10 mM phosphate buffer (pH 7.2) containing 1 mM MgCl_2_, 80 µM cytochrome C, and 2 mM sodium azide. A total of 0.2 mM NADPH was included in the reaction mixture, after which the absorbance was measured at 550 nm [24].

### 2.12. Two-Dimensional Gel Electrophoresis of S. cervi Protein Samples After EOS Treatment

#### 2.12.1. Sample Preparation

The homogenate of treated and control samples, in two different sets, was combined with ten times its volume of ice-cold acetone and incubated at −20 °C for 3 h. The protein was precipitated via centrifugation at 5500× *g* for 10 min at 4 °C. The pellet was subsequently dissolved in rehydration solution (40 mM Tris-HCl pH 7.5, 7 M Urea, 2 M Thiourea, 2% CHAPS, 15 mM DTT, 0.5% *v*/*v* IPG buffer, 1 mM PMSF) and subjected to isoelectric focusing (IEF).

#### 2.12.2. Two-Dimensional Electrophoresis

Initially, pre-cast ReadyStrips^TM^ (BioRad), 11 cm with a pH range of 3–10, were rehydrated using a solution containing 400 µg of protein and 0.2% bromophenol blue. Next, strips were immersed in solution containing pharmalyte and incubated for 10 h at 25 °C. Further, isoelectric focusing (IEF) was conducted using the PROTEAN IEF cell (BioRad) at 250 V (linear gradient) for 25 min, 8000 V (linear gradient) for 2.5 h, 8000 V (Rapid) for 7 h, and 500 V used for storage. Subsequently, IEF strips were incubated in equilibration buffer containing DTT and iodoacetamide for reduction and alkylation, respectively (50 mM Tris-HCl pH 8.8, 6 M Urea, 30% (*v*/*v*) glycerol, 2% (*v*/*v*) SDS, 1% DTT or 2.5% iodoacetamide) for 15 min for each. The strips were further loaded and subjected to a 12.5% SDS-PAGE [25]. The proteins were separated via electrophoresis, and the gel was stained with colloidal Coomassie solution (0.12% Coomassie brilliant blue G 250, 10% (NH_4_)_2_SO_4_, 10% orthophosphoric acid, and 20% methanol).

### 2.13. Image Analysis and Quantitation

The PDQuest image analysis software basic version 8.0.1 was utilized for the quantitative examination of protein spots. Two independent experiments were undertaken, and a scatter plot signed ranks test was executed to assess the statistically significant differences among protein spots. Protein spot volume intensity with changes greater than 1.5-fold were only documented as significantly increased or decreased.

### 2.14. Reduction and Trypsin Digestion of Differentially Expressed Spots from 2D Gels

The differentially expressed protein spots were washed several times with water after being manually excised. A total of 15 mM K_3_[Fe (CN)_6_] and 50 mM NH_4_HCO_3_ in 1:1 ratio was used for destaining of the Coomassie brilliant blue-stained spots via gentle shaking for 10 min. This step was repeated again till the blue color was not visible. Dehydration was achieved by covering the gel pieces with 100% acetonitrile (ACN) for 30 min at 4 °C. A total of 10 mM DDT in 100 mM of NH_4_HCO_3_ was added to the gel pieces, and reduction was carried out for 45 min. The accesses liquid was removed, and gel pieces were washed with 100% ACN and air dried for 10–15 min. For trypsin digestion, gel spots were hydrated with 50 µL of 0.01 mg/mL trypsin (Sigma) solution for 30 min at 4 °C. The samples were incubated overnight at 37 °C; the next day, formic acid was added to a final concentration of 5%, followed by the addition of 100% ACN. The supernatant was pipetted into a new vial, and two more extractions with gel pieces were performed using 1:1 ACN/water and 1% formic acid in ACN. The digested samples were desalted utilizing C18 cartridges and completely dried using SpeedVac. The peptides were ultimately preserved at −80 °C for further proteomics investigation.

### 2.15. High-Resolution Accurate Mass Spectrometry (HRAMS) Analysis

The samples underwent analysis using the high-resolution accurate mass spectrometer at the Central Discovery Centre, Banaras Hindu University, for protein identification. A peptide analysis was conducted using an Orbitrap Eclipse Tribrid mass spectrometer equipped with nano LC and UHPLC (Thermo Fisher Scientific, Waltham, MA, USA). A total of 1 μg of the digested and desalted peptide mixture was introduced into a reverse-phase C_18_ trap column (Acclaim PepMap100; 75 μm × 2 cm, nanoViper) linked to a reverse-phase C_18_ analytical column (PepMapTM RSLC C_18_; 2 μm, 100 Å, 75 μm × 50 cm) using buffer A (2% acetonitrile in water + 0.1% formic acid). Peptides were separated at a flow rate of 300 nL/min utilizing a linear gradient of buffer B (85% acetonitrile in water + 0.1% formic acid) as follows: 0 to 5 min—2% of buffer B; 5 to 30 min—2–20% of buffer B for; 30 to 85 min—20–50% of buffer B; 85 to100 min—50–100% buffer B; 101 to 113 min—100% of buffer B; followed by 2% buffer B to 120 min. Comprehensive mass spectrometry scans were conducted within the range of 200–1600 *m*/*z*, utilizing Thermo Scientific™ Proteome Discoverer™ software v 3.0 (Thermo Fisher Scientific) for peptide identification. The MS/MS spectra of individual peptides were aligned with the database sequence using Proteome Discoverer™ software.

### 2.16. Statistical Analysis

All in vitro tests were conducted in triplicate (n = 3) and repeated thrice unless stated otherwise. The data are shown as mean ± SD, derived using Graph Pad Prism 8.1 software (GraphPad Software, La Jolla, CA). The statistical significance between control and EOS-treated parasites was determined using Student’s *t*-test. Statistical significance was determined as * *p* < 0.05, ** *p* < 0.01, and *** *p* < 0.001.

### 2.17. String Analysis of Differentially Expressed Proteins

Protein–protein interaction network analysis was used to forecast the functional connections of differentially expressed proteins (DEPs) upon EOS treatment. The STRING tool v12.0 was utilized, and the interaction format was designated exclusively for “*Brugia malayi*” (string-db.org accessed on 15 March 2022). The gene ontology terms, biological processes, and pathway analyses for each dataset were generated using STRING program [26].

### 2.18. Retrieval of Targeted Protein Structures

The three-dimensional structure of *Wuchereria bancrofti* Glyceraldehyde 3-phosphate dehydrogenase (GAPDH) (4K9D, 10.2210/pdb4K9D/pdb) was obtained from the Protein Data Bank [27]. The three-dimensional structures of filarial heat shock protein 70 (HSP70), adenylate Kinase (ADK), enolase, and phosphoglycerate kinase (PGK) were absent from the PDB databases. Consequently, the template sequences of *W. bancrofti* HSP 70 (accession no. AAF32254.1), ADK (accession no. EJW76745.1), enolase (accession no. AHI18146.1), and PGK (accession no. EJW85467.1) were chosen for protein structure modeling from the NCBI database based on optimal score and query coverage. LOMETS, a meta server methodology for template-based protein structure prediction, was utilized for retrieving the three-dimensional structures of proteins.

### 2.19. Protein Model Validation

The validation of protein models was conducted using the SAVES v6.0 system (saves.mbi.ucla.edu accessed on 20 March 2022). The quality evaluation and hydrogen bond parameters of the HSP70, ADK, enolase, GAPDH, and PGK models were assessed using the VADAR 1.8 server [28]. The active sites of the selected 3D structures of HSP70, ADK, enolase, GAPDH, and PGK were predicted using Discovery Studio 3.5.

### 2.20. Retrieval of Ligand

The compounds eugenol, kaempferol, luteolin, rosmarinic acid, rutin, and ursolic acid, identified as the principal bioactive constituents of OS (Mishra et al., 2022 [12]), were chosen for docking study. Ligand structures were obtained from the PubChem Compound Database [29] in SDF format and subsequently converted to PDB format using Discovery Studio 3.5. The Lipinski filter [30,31] and admetSAR software v2.0 [32] were employed to predict drug-like properties.

### 2.21. Molecular Docking

Docking study was conducted using the principal differentially expressed proteins of filarial parasites, namely, HSP70, ADK, enolase, GAPDH, and PGK, against the bioactive constituents of *Ocimum sanctum* (OSBCs), specifically, eugenol, kaempferol, luteolin, rosmarinic acid, rutin, and ursolic acid. The PatchDock server version Beta 1.3 and YASARA tools version 15.11.18 were employed to achieve the optimal docking configuration of OSBC with the differentially expressed proteins [33]. Visualization of complexes was performed using Biovia Discovery Studio 3.5.

## 3. Results

### 3.1. Viability

Adult female *S. cervi* worms were incubated in KRB maintenance medium for 6 h at 37 °C and 5% CO_2_ with various concentrations of EOS. The EOS treatment was given at 125, 250, and 375 µg/mL concentrations, and the viability of *S. cervi* parasites was assessed over the entire period of 6 h by MTT assay. A gradual reduction in the viability was observed with increasing concentration of EOS, and maximum reduction (90.84%) was observed at 375 µg/mL concentration (Figure 1A). The LC_50_ value of EOS was determined using Origin pro 2024 software and was calculated to be 197.24 µg/mL.

### 3.2. Effect of EOS on Oxidative Stress Markers

The effect of EOS on adult female *S. cervi* was observed after 6 h of treatment. The glutathione (GSH) level was decreased (39.23%, *p* = 0.025) in worms treated with 375 µg/mL EOS. The GST activity was significantly diminished in treated parasites relative to control worms after 6 h of incubation. It was observed that the GST activity of EOS-treated *S. cervi* decreased by 11.02% (125 µg/mL), 31.93% (250 µg/mL), and 50.48% (375 µg/mL) after 6 h of EOS incubation. A dose-dependent inhibition in enzymatic activity of GR was also seen. A reduction of 25.78% in 125 µg/mL, 40.44% in 250 µg/mL, and 60.1% in 375 µg/mL EOS-treated *S. cervi*, was observed. Thioredoxin reductase activity was reduced by 37.36%, 67.88%, and 77.98% in EOS-treated parasites at EOS concentrations of 125 µg/mL, 250 µg/mL, and 375 µg/mL, respectively (Table 1). The protein oxidation following EOS treatment was measured by quantitating the protein carbonyl content. Significant increments of 21.68% (125 µg/mL), 34.98% (250 µg/mL), and 48.99% (375 µg/mL) in the protein carbonyl content of EOS-treated worms as compared with control worms was observed (Figure 1B). Lipid peroxidation, as indicated by MDA levels, was assessed in control and EOS-treated parasites. The MDA level was significantly elevated to 14.58%, 36.25%, and 51.92% in parasites subjected to 125, 250, and 375 µg/mL compared to the control worms (Figure 1C). Our experimental evidence showed that EOS treatment led to an increase in NADPH oxidase activity by 26.66%, 60.00%, and 88.88% after treatment with 125, 250, and 375 µg/mL EOS extract, respectively (Figure 1D).

### 3.3. Proteforms Profile of S. cervi After EOS Treatment

The observation of viability and oxidative stress markers confirmed that *S. cervi* parasites can tolerate EOS concentrations of 250 µg/mL up to 6 h. Hence, this concentration was chosen for analyzing the alteration in the proteforms profile of filarial parasites. The adult female *S. cervi* worms exposed to 250 µg/mL EOS exhibited notable changes in their proteforms profile relative to the control worms (Figure 2). In 2D gel, a total of 122 spots were observed in the control and 102 spots in EOS-treated parasites, respectively. PD quest analysis showed that 20 decreased and 11 increased protein spots were present in EOS-treated worms. The Pearson correlation coefficient between the treated and control samples was observed at 0.647 (Appendix A). The differentially expressed spots were further identified via HRAMS analysis (Table 2). The identified proteins spots that were increased were mostly stress-responsive proteins and chaperons like heat-shock protein 70 (HSP 70), as well as thioredoxin domain containing protein-3 homolog. The decreased proteins were adenylate kinase, enolase, glyceraldehyde 3 phosphate dehydrogenase, and phosphoglycerate kinase, which were mostly related to energy metabolism.

### 3.4. Protein Networks and Functional Analysis

Among the 16 DEPs, 9 proteins exhibited interrelations, 1 protein was isolated with no connections to any nodes, and 7 proteins were absent from the STRING database (Figure 3). The interaction network showed a local clustering coefficient of 0.455 and a PPI enrichment *p*-value of 7.11 × 10^−6^. Functional pathway analysis of nine DEPs was investigated via a biological process and KEGG. The nine interrelated DEPs were involved in eight biological processes, including nucleoside diphosphate phosphorylation (17%), glycolytic process (15%), ATP metabolism process (17%), glucose metabolic process (9%), organic substance catabolic process (18%), cellular response to heat (6%), gluconeogenesis (6%), protein refolding (6%), and chaperone cofactor-dependent protein refolding (6%). The KEGG pathways study indicated that the major DEPs were associated with glycolysis/gluconeogenesis, amino acid biosynthesis pathways, carbon metabolism, and metabolic processes.

### 3.5. Target Protein Retrieval and Validation

The major differentially expressed proteins (DEPs) identified in the 2D proteome analysis of EOS-treated *S. cervi* parasites, i.e., ADK, enolase, GAPDH, HSP 70, and PGK, were chosen as drug targets for in silico study. Based on earlier reports, major OSBC like eugenol, kaempferol, luteolin, rosmarinic acid, rutin, and ursolic acid were selected for molecular docking against *S. cervi* DEPs (Pandey et al., 2015; Mishra et al., 2022 [10,12]). The 3D structures of ADK, enolase, HSP70, and PGK were modeled by the LOMET server, and the GAPDH protein structure was retrieved from PDB. The structural properties and H-bond statistics of all five targeted proteins were analyzed via a Vadar 1.8 server (Appendix A). The Ramachandran plot for stereochemical quality of targeted proteins was checked by the Procheck server available on SAVES v6.0 (Figure 4). The ψ (psi) and φ (phi) distribution of the ADK structure showed 96.6% amino acid (a.a.) residues in the core region and 2.8% residues in the allowed region. Similarly, 86.4% a.a. of enolase were in the core region, and 11.8% were in the allowed region. The structures of GAPDH and HSP 70 had 93.5% and 90.4% a.a. residues in the core region, whereas they had 6.1% and 8.3% in the allowed region, respectively. PGK has 86.9% in the core region and 12.6% in the allowed region. (Appendix A).

### 3.6. Molecular Docking Analysis

The structures of OSBC eugenol, kaempferol, luteolin, rosmarinic acid, rutin, and Ursolic acid were retrieved from the PubChem database (Appendix A). Molecular docking of the aforementioned OSBCs and *S. cervi* DEPs was via by YASARA software version 15.11.18 and the PatchDock server version Beta 1.3 (Table 3). The amino acid interactions of the docked complexes were visualized by Discovery studio 3.5 (Figure 5). The highest scoring OSBC in molecular docking were rutin, luteolin, and ursolic acid. Among these, rutin formed the maximum number of hydrogen bonds with GAPDH (7 bonds) and enolase (7 bonds), whereas rosmarinic acid formed the maximum number of hydrogen bonds with ADK (5 bonds) and PGK (11 bonds). Eugenol formed hydrogen bonds only with HSP 70 (one bond) and did not form any hydrogen bonds with other DEPs. The anti-filarial drugs, DEC and albendazole, formed hydrogen bonds with PGK (Albendazole-1, DEC-2) and ADK (DEC-3) (Appendix A). Rutin showed the highest binding energy with ADK (9.0130 Kcal/mol), enolase (8.7070 Kcal/mol), and HSP 70 (7.9120 Kcal/mol). Ursolic acid had the strongest interaction with GAPDH (9.2360 Kcal/mol) and PGK (10.2220 Kcal/mol). The GSC score and AI area, as predicted by the PatchDock server version Beta 1.3, were highest for rutin and ursolic acid. The OSBC eugenol had the lowest binding energy and dissociation constant against *S. cervi* DEPs. The drug-likeness properties and ADME prediction of OSBCs used in this study is based upon a previous report from the laboratory (Mishra et al., 2022 [12]).

## 4. Discussion

Modern drugs used to treat infectious diseases are either directly or indirectly derived from the bioactive components of plants [34]. In the past few decades, traditional medicinal herbs and phytochemicals have gained great interest for the development of innovative and effective pharmaceuticals with fewer side effects than allopathic treatments. Additionally, the intake of dietary phytochemicals as chemoprevention agents for the detoxification of reactive species, and as nutraceuticals, is becoming popular even in developed countries of Europe and America too [35]. Herbal medicines are a time-tested and cost-effective alternative therapy for developing countries, where huge populations are living at the risk of LF infections. Numerous in vitro and in vivo studies have already demonstrated that crude extracts from plants, vital oils, and extracted active ingredients/compounds possess anti-filarial activity, which can be scientifically explored to discover new anti-filarial lead compounds [36].

The therapeutic potential of OS is immense, and the herb is already much explored as a potent antibacterial, antiviral, cardio-, hepato-, and nephro-protective [8]. OS has diverse medicinal properties and has pharmacological activities against lung cancer, malaria, Leishmaniasis, HIV, neurodegenerative, and lung diseases [37,38,39,40,41,42]. The ethanolic extract of OS can induce apoptosis in lung carcinoma, and its essential oil promotes cytotoxicity and apoptosis in colorectal adenocarcinoma [43,44]. Several OS formulations, like aqueous, ethanolic, and methanolic leaf extracts, as well as seed oil and fresh leaf paste, have been tested for their therapeutic potential [8].

In our study, the treatment of worms with ethanolic extract of *O. sanctum* (EOS) reduced the viability of *S. cervi* parasites in a dose- and time-dependent manner. It has been shown previously that anti-filarial medication—DEC, albendazole, and Ivermectin—do does not affect the viability of adult *S. cervi*, even after exposures for 4–8 h [45,46]. Nevertheless, the EOS showed significant macrofilaricidal activity within 6 h of treatment, as assessed via the MTT reduction assay. The reported LC_50_ values of some other herbal crude extracts tested against adult female *S. cervi* were 945.4 µg/mL for *Andrographis paniculata* and 47.12 µg/mL for *Azadirachta indica* [13,47]. The LC_50_ value of EOS is 197.24 µg/mL, which is much lower than the reported LC_50_ value of *A. paniculata* crude extract. Although the LC_50_ of EOS was higher than *A. indica* extract, the exposure of worms to *A. indica* was for 24 h, which is much longer than the EOS exposure in this study. It was observed that the EOS was more potent than the aqueous OS leaf extract (unpublished data). Earlier OS extract was reported as ovicidal and larvicidal in gastrointestinal nematodes, though the mechanism of anti-helminthic action of OS remained unclear [48]. As a result, the current study attempts to analyze the mechanism of anti-filarial action of OS and its bioactive components.

The filarial parasites survive easily for several years by successfully evading the host’s immune attacks. The parasites employ a battery of enzymatic and non-enzymatic defense proteins/enzymes such as glutathione-S-transferase, glutathione (GSH), GR, and NADPH oxidase as their defense system [49]. GSH regulates cellular homeostasis via the oxidation of SH groups of proteins involved in various functions and is also important as a scavenger of free radicals, thus maintaining a reducing environment within the cells. Additionally, equally important are the enzymes GST and GR, which act as free radicals scavengers, hence protecting the parasites from the hosts’ immune attacks [50]. A significant decrease in the free glutathione levels of EOS-treated *S. cervi* parasites were observed after exposure to 375 µg/mL of EOS (−39.23%). The antioxidant enzymes GST and GR showed similar patterns of decrease in enzymatic activity following EOS treatment. The effect of EOS on the antioxidant defense mechanism of the filarial parasites was pronounced and could be a major factor contributing to its anti-filarial activity.

The disruption of the antioxidant defense mechanism of *S. cervi* parasites prompted an investigation of oxidative stress parameters like lipid peroxidation, NADPH oxidase activity, and protein carbonyl content. An increase in protein carbonyl content, lipid peroxidation, and NADPH oxidase activity was observed after EOS exposure. An increase of almost 52% in levels of lipid peroxidation strongly indicates alteration in membrane permeability and fluidity, leading to irreversible membrane damage [51]. The NADPH oxidase has been implicated in varied functions, such as cellular signaling, regulation of gene expression, and host defense against viral and bacterial infections. An increase in NADPH oxidase activity leads to the transport of electrons to oxygen in the extracellular environment, resulting in the formation of superoxide anions [52]. The superoxide anion, so generated, augments the levels of hydrogen peroxide and other reactive oxygen species in the extracellular milieu of the filarial parasites. Once formed, the reactive oxygen species promote varieties of oxidative damage, such as the oxidation of proteins, which is estimated by measuring the increase in the protein carbonyl content [53]. The protein carbonyl content of the EOS-treated parasites was significantly higher than the control *S. cervi* worms (*p* = 0.0043), which could be due to the generation of ROS and oxidative stress.

The proteoforms profile of *S. cervi* adult female parasites after EOS treatment (250 µg/mL) exhibited a reduction in the number of protein spots as compared to the control worms. The gel analysis report showed that the differentially expressed proteins participated in several biological processes, including drug detoxification, redox regulation, energy metabolism, stress response, signaling, and structural and cytoskeleton. An increase in the levels of stress response and heat-shock proteins like HSP 70 and HSP 18 was also observed after EOS treatment. The molecular chaperone HSP 70 is an efficient redox scavenger and is overexpressed under stress conditions like hypoxia, nutrient depletion, and oxidative stress [54]. The sentinel heat shock protein, HSP 70, protects the cells from different stresses and imbalances that may occur due to aging or pathophysiological changes [55]. It is quite possible that the *S. cervi* parasites perceived EOS exposure as a stress stimulus and responded by overexpressing the heat-shock proteins HSP 70 and HSP 18.

Additionally, EOS treatment had a significant effect on the enzymes of energy metabolism like enolase and GAPDH, which were decreased in the treated parasites as compared to control worms. These enzymes are involved in the glycolytic pathway, and a decrease in the expression can be correlated with the decrease in the energy generation [56] during EOS exposure. ADK was significantly decreased in the treated parasites, which would impact the AMP-driven metabolic signaling, intracellular energetic communication, as well as the ATP supply [57]. The antioxidant proteins thioredoxin domain containing protein-3 homolog was increased, while GST was decreased following exposure to EOS. Decreased in GST in filarial parasites is associated with the loss of metabolic homeostasis, culminating in parasites’ death [58]. The EOS-treated *S. cervi* worms also showed diminished levels of the p27 molecule, which is an inhibitor of CDK (cyclin dependent kinases). p27 is associated with cell viability, and its reduced expression could render the parasites vulnerable to death [53].

The in silico approach offers a considerable advantage over in vivo lab research in terms of time and resource involvement. The effectiveness of molecular docking in the realms of drug discovery and design is now widely recognized. In particular, the molecular targets of natural compounds used in disease management are initially verified through the use of molecular docking. In natural compounds/nutraceutical research, molecular docking studies are employed to provide crucial information prior to in vitro experiments. In order to identify the anti-filarial effect of *O. sanctum* bioactive compounds, molecular docking was performed. The *O. sanctum* bioactive compounds used in this report are either reported as major components of OS leaf extract in the literature or have proven pharmacological and therapeutics activities. The computational evaluation of the ligand–receptor electrostatics can be predicted via docking studies [59]. The molecular docking technique predicts the orientation of the ligand in a complex that it may form with proteins or enzymes. The form and structure of the docked complex, as well as the Van der Waal forces, ionic bonds, and hydrogen bonds between ligand and receptor, are taken into account to quantify the molecular interactions [60].

The three best bioactive compounds based on molecular docking analysis were ursolic acid, rutin, and rosmarinic acid. Ursolic acid, a natural pentacyclic triterpenoid carboxylic acid, has proven anti-filarial activity against *B. malayi* and *S. cervi* microfilariae and adult parasites [61,62]. In our investigation, ursolic acid has given high binding energies against the differentially expressed proteins, ADK, enolase, GAPDH, HSP 70, and PGK. Another in silico study has also reported the high binding affinity of ursolic acid (UA) with *Brugia malayi* GST; hence, UA may be considered as a potential lead compound for designing novel anti-filarial compounds/drugs [61].

The second-best OSBC with anti-filarial activity identified via molecular docking is the flavonoid rutin. It has a wide range of pharmaceutical benefits, including antibacterial, anticancer, antifungal, antiretroviral, and antimalarial activities [63]. In pancreatic cancer cells, rutin has been shown to promote apoptosis by upregulating miR-877-3p expression, which, in turn, represses bcl-2 transcription [64]. However, the anti-filarial effect of rutin has not been much explored, except in a solitary report on *B. malayi* worms, where it was shown to have macro and microfilaricidal activities [65]. In molecular docking, rutin showed high binding energies and low dissociation constant with ADK, enolase, GAPDH, HSP 70, and PGK. Based on the earlier findings and our work, the bioactive compound rutin can certainly be explored further for its anti-filarial effects. The phenolic compound rosmarinic acid also showed good docking scores against the targeted *S. cervi* DEPs. Rosmarinic acid is an anticancer, antiangiogenic, anti-inflammatory, and antimicrobial agent [66]. We observed that rosmarinic acid could bind strongly to several proteins, such as ADK, enolase, GAPDH, HSP 70, and PGK. The antiparasitic and anti-filarial activities of rosmarinic acid have not been investigated so far, and more experimental validations can delineate its mode of anti-filarial action.

Although the study was meticulously planned, the present investigation still has some limitations. Despite the substantial benefits of our 2D profiling, it has a drawback in terms of throughput; on average, a 2D study takes around two–three days. Also, solubilization is a major concern because the hydrophobic and membrane-bound proteins in the worm tegument need to be solubilized separately using detergents; hence, the proteoform profile of this research does not include the impact of EOS on these proteoforms. Additionally, the alterations in proteomes were quantified using 2D image analysis tools, which required a number of replicates for comparison and spot matching [67] since we have used protease inhibitors such as PMSF and EDTA, which may be a reason for the loss of certain proteins and/or their absence from the 2D PAGE gels.

Thus, in essence, this work proves the macrofilaricidal effect of OS by mediating its action on filarial antioxidant enzymes/proteins, lipid peroxidation, and protein carbonylation. Following this, the proteoforms analysis successfully identified putative filarial drug targets that were validated in silico via molecular docking against OS bioactive compounds.

## 5. Conclusions

The ex vivo experiments demonstrated that EOS had significantly affected the viability and biochemical markers of adult female *S. cervi* parasites. The effect of EOS was mainly due to the increase in reactive oxygen production evidenced by the decrease in GST, GR, and GSH levels. Proteomic analysis and biochemical assays confirmed that EOS could induce apoptosis in *S. cervi*. Molecular docking with OS bioactive compounds proved that eugenol, kaempferol, luteolin, rosmarinic acid, rutin, and ursolic acid could stably interact with DEPs of *S. cervi*. In conclusion, the overall outcomes from our study provide insight into the effect of OS and its bioactive compounds on *S. cervi*. Further studies on the multi-inhibitory potential of OS and its derivatives can lead to the development of better anthelminthic drugs in future.

## Figures and Tables

**Figure 1 proteomes-13-00002-f001:**
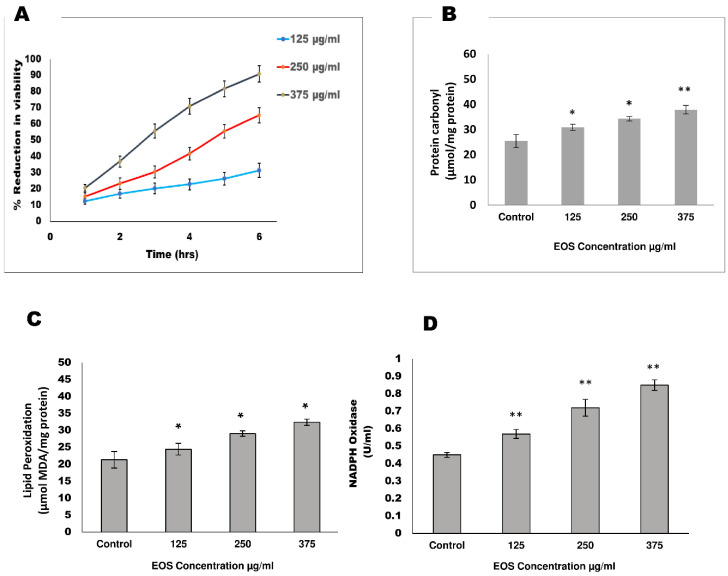
Effect of EOS on the adult female *S. cervi* worms. Adult worms (n = 5) of equal size were exposed at 125 µg/mL, 250 µg/mL, and 375 µg/mL concentration of ethanolic extract of OS in 20 mL of KRB maintenance medium at 37 °C. (**A**) The viability of parasites was determined via MTT assay after 6 h of incubation; percent reduction in viability was calculated with respect to the control group of *S. cervi* worms. (**B**) Protein carbonyl content is expressed in terms of μmol/mg protein. (**C**) Lipid peroxidation level is expressed in terms of μmol MDA/mg protein. (**D**) Activity of the enzyme NADPH oxidase activity is expressed as U/mL. Data expressed are the mean ± SD of at least three values (n = 3). *p* values < 0.05 (*), <0.01 (**) were considered as significant statistically.

**Figure 2 proteomes-13-00002-f002:**
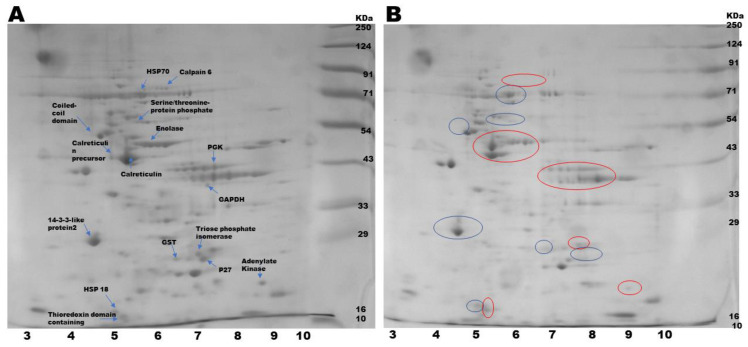
Two-dimensional electrophoresis profile of Coomassie-stained crude extract of *S. cervi* exposed to EOS: (**A**) control; (**B**) treated. The first dimension was performed on pH 3–10 IPG strips followed by a second dimension on 10% SDS-PAGE. Increased protein spots are indicated by blue circles; decreased spots are indicated by red circles.

**Figure 3 proteomes-13-00002-f003:**
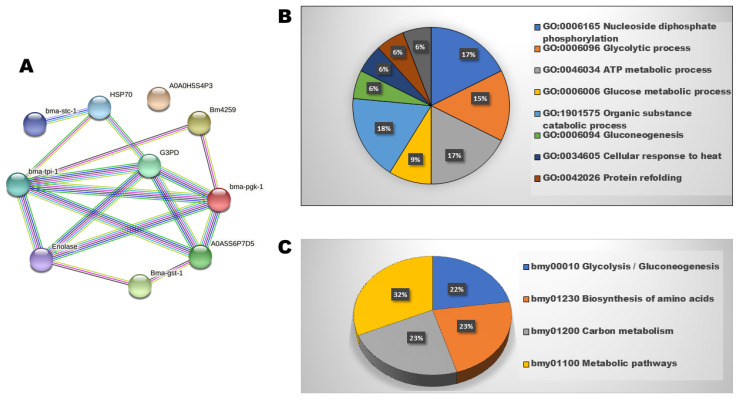
Interaction network and functional analysis of identified proteins of *S. cervi* DEPs after EOS treatment: (**A**) interaction network depicting the association among the differentially regulated proteins; (**B**) gene ontology (GO) biological process, represented in the pie chart as a percentage; (**C**) KEGG pathways, represented in the pie chart as a percentage.

**Figure 4 proteomes-13-00002-f004:**
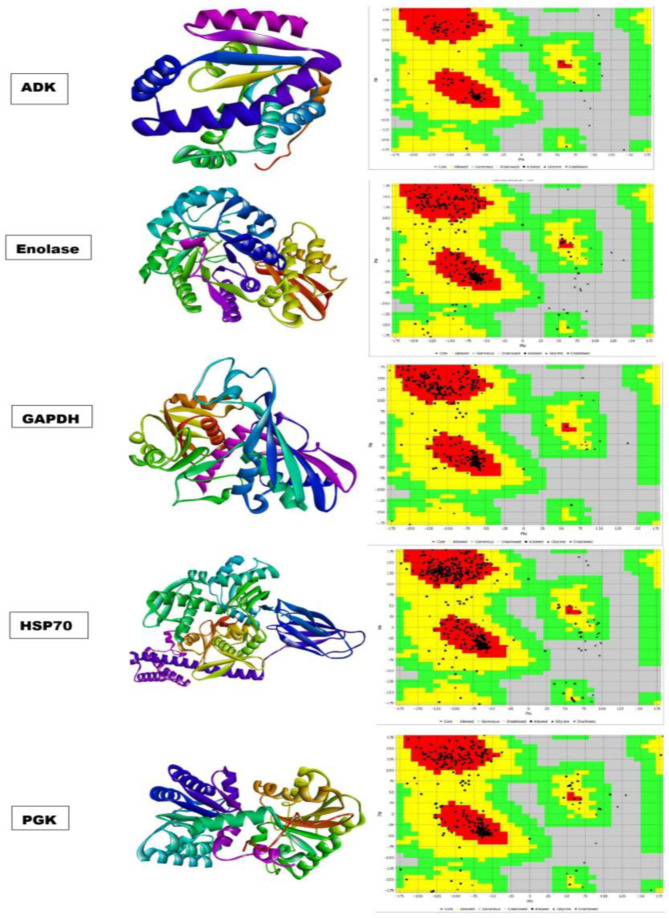
Three-dimensional structure and quality validation, via Procheck server, of differentially expressed proteins (DEPs) used in molecular docking analysis. Square shows the Ramachandran plot of respective protein red color shows core region, yellow shows allowed region, green color shows generous region and gray color showed disallowed region.

**Figure 5 proteomes-13-00002-f005:**
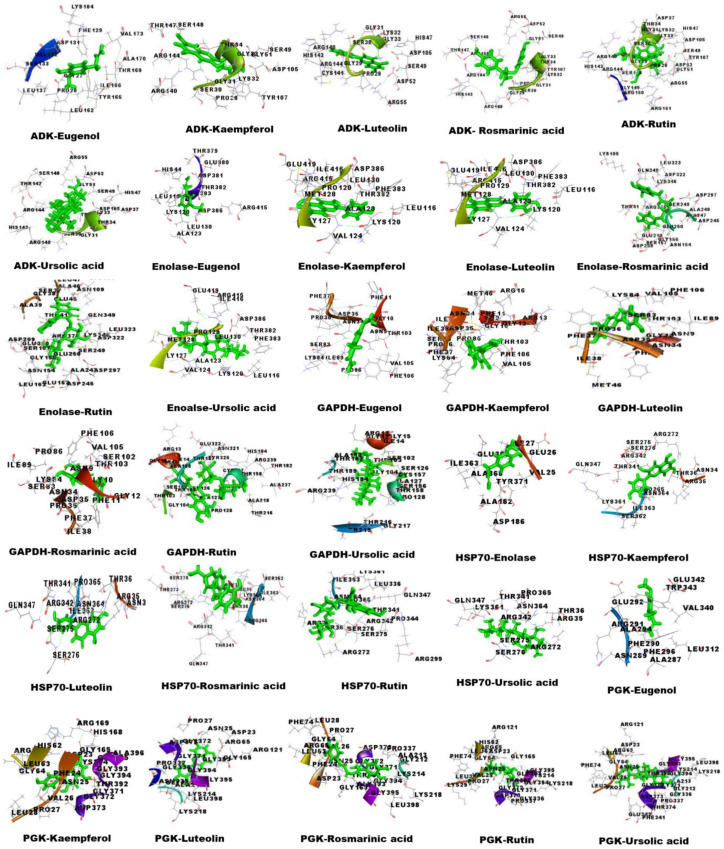
Three-dimensional interactions of *S. cervi* DEPs with OS bioactive compounds. The docked complexes were visualized, and the ligands are represented in green color. Interacting residues are as labeled.

**Table 1 proteomes-13-00002-t001:** Effect of ethanolic extract of *Ocimum sanctum* on *S. cervi* antioxidant stress markers.

Samples	GSH (µM/mg Protein)	GST (U/mL)	GR (U/mL)	TRxR
Control	6.50 ± 0.120	22.86 ± 0.568	16.02 ± 1.110	18.12 ± 0.832
125 µg/mL	5.82 ± 0.513	20.34 ± 0.862 *	11.89 ± 0.636 **	11.35 ± 1.232 **
	(−10.46%)	(−11.02%)	(−25.78%)	(−37.36%)
250 µg/mL	4.24 ± 0.982	15.56 ± 1.101 *	9.54 ± 0.350 **	5.82 ± 0.345 ***
	(−34.76%)	(−31.93%)	(−40.44%)	(−67.88%)
375 µg/mL	3.95 ± 0.832 *	11.32 ± 1.926 *	6.38 ± 0.832 **	3.99 ± 0.127 ***
	(−39.23%)	(−50.48%)	(−60.17%)	(−77.98%)

The enzyme activity GSH, GST, GR, and TRxR were assessed in the control and EOS-treated parasites after 6 h of incubation. The values in parentheses indicate percentage activation/inhibition in comparison to control. Data expressed are the mean ± SD of at least three values (n = 3). *p*-value < 0.05 considered significant. *** *p* < 0.001, ** *p* < 0.01, * *p* < 0.05.

**Table 2 proteomes-13-00002-t002:** HRAMS analysis of differentially expressed proteins of *S. cervi*.

S.N.	Protein ^a^ (Accession Number)	Function	MW/pI		Identified Peptide	Fold Change ^b,c^
			Expected	Experimental		
Cytosol stress response protein/chaperons						
**1**	Heat-shock protein 70 [P27541]	Act as molecular chaperone; help in protein folding, transport, and assembly; and protect cell against stress	72/5.6	70.2/5.47	[K].MKETAEAFLGHAVK.[D] [RH].NVLIFDLGGGTFDVSILTIEDGIFEVK.[S] [R].IINEPTAAAIAYGLDK.[K]	+1.9 *
**2**	p27 [A0A4E9ERK7]	Cell cycle regulation	25.86/7.9	24.1/6.32	[K].IDVTPSNYSVLDTEFGSMR.[E] [R].AVFREYNQEFMLPR.[G]	−2.4 **
Cytosol energy metabolism proteins						
**3**	Adenylate kinase isoenzyme 1 [J9AQV1]	Catalyzes phosphoryl transferase, having a role in metabolic monitoring and AMP signaling	22.95/8.8	22.8/8.34	[R].LHTYITATAPVVDYYQK.[Q] [K].YGLTHLSSGDLLRAEVK.[S] [K].ANVPIFFIVGGPGSGKGTQCDKIVAK.[Y]	−3.6 ***
**4**	Enolase [Q5GTG4]	Role in glycolysis and gluconeogenesis	47/5.9	46.4/5.67	[R].LAKYNELIR.[I]	−3.3 ***
**5**	Glyceraldehyde−3-phosphate dehydrogenase [A0A4E9FA01]	Role in energy metabolism	36/7.1	36.1/7.84	[K].LTGMAFRVPTPDVSVVDLTCR.[L] [R].VPTPDVSVVDLTCR.[L] [K].AVGKVIPDLNGKLTGMAFR.[V]	−3.3 ***
**6**	Phosphoglycerate kinase [A0A4E9EYJ5]	Glycolytic enzyme	44/7.68	44.6/7.88	[K].MEFTLEPVAAELK.[A] [R].AKTIVWNGPAGVFEWENFSK.[G] [R].KMEFTLEPVAAELK.[A]	−3.2 ***
Antioxidant protein/enzymes						
**7**	Glutathione S-transferase [E3UV59]	Antioxidant enzyme; detoxification of endogenous and xenobiotics compounds	25/5.88	24.1/6.68	[K].DILPVELAKFEK.[L] [K].FEKLLATR.[D]	−2.6 **
**8**	Thioredoxin domain-containing protein [A0A4E9FJK0]	Antioxidant enzyme	22.1/7.09	22.1/7.06	[R].LIQAFQFVDKHGEVCPANWHPGSETIKPGVK.[E] [K].GKYVVLFFYPLDFTFVCPTEIIAFSDR.[I]	+1.5 *
Signaling protein						
**9**	Coiled-coil domain-containing protein 6 [A0A1I8E9M6]	Structural motifs involved in a variety of important interactions	60.92/5.19	60.9/5.26	[R].AFAASETTRENDEDNCMAALLNR.[M]	+2.4 **
Protein Digestion and folding protein						
**10**	Calreticulin precursor [A0A0J9XSV8]	Calcium-binding chaperone role in transcription regulation	47.42/4.78	49.4/4.87	[K].KVHVIFHYKGR.[N] [K].HKDDFGKWEISHGK.[F]	−3.5 ***

^a^ Matched proteins were retrieved from the Uniprot database (http//:www.uniprot.org, accessed on 11 January 2022). ^b^ Fold change calculated from mean intensity and volume of spots observed in treated and control groups run on two different sets of experiments. ^c^ Student’s *t* test were used for calculating the statistical difference between control and treated groups. *p*-value < 0.05 considered as significant. *** *p* < 0.001, ** *p* < 0.01, * *p* < 0.05.

**Table 3 proteomes-13-00002-t003:** Docking summary of treated *S. cervi* differentially expressed proteins with bioactive compounds of *Ocimum sanctum*.

Receptor	Name of Ligand	Binding Energy (Kcal/mol)	Dissociation Constant (µm)	GSC Score	AI Area
**ADK**	Eugenol	5.7090	65.3502	3286	381.30
	Kaempferol	7.4400	3.5189	4244	473.50
	Luteolin	7.5920	2.7226	4280	497.40
	Rosmarinic acid	7.9160	1.5757	5100	636.50
	Rutin	9.0130	0.2473	5614	630.70
	Ursolic acid	7.6660	2.4029	6176	791.80
	Albendazole	5.9560	43.0719	4426	518.70
	DEC	5.0650	193.7785	4038	475.00
**Enolase**	Eugenol	5.4220	106.0770	3484	388.50
	Kaempferol	8.0820	1.1907	4062	388.50
	Luteolin	7.630	2.4252	3760	438.70
	Rosmarinic acid	7.6640	2.4110	4634	540.60
	Rutin	8.7070	0.4146	5488	711.60
	Ursolic acid	8.0270	1.3065	5800	699.80
	Albendazole	5.6240	75.4316	3994	480.80
	DEC	4.9930	218.8177	3764	421.30
**GAPDH**	Eugenol	6.1610	30.4738	3124	331.50
	Kaempferol	7.5610	2.8688	3664	406.20
	Luteolin	7.9720	1.4336	3710	426.50
	Rosmarinic acid	7.6930	2.2959	4252	494.10
	Rutin	8.5540	0.5368	5538	651.60
	Ursolic acid	9.2360	0.1697	5352	578.60
	Albendazole	5.8850	48.5554	3730	451.10
	DEC	5.0210	208.7171	3500	372.30
**HSP70**	Eugenol	4.8890	260.8047	3232	3555.30
	Kaempferol	6.6860	12.5631	3914	441.00
	Luteolin	7.0660	6.6153	3888	417.10
	Rosmarinic acid	7.2870	4.5557	4590	548.30
	Rutin	7.9120	1.5864	5268	719.70
	Ursolic acid	7.8070	1.8940	5276	658.40
	Albendazole	6.0360	37.6316	3830	438.10
	DEC	4.8560	275.7432	3676	418.90
**PGK**	Eugenol	5.5360	7.5101	3162	335.40
	Kaempferol	8.0840	1.1872	3546	373.70
	Luteolin	8.4230	0.6696	3528	390.70
	Rosmarinic acid	8.6300	0.4722	4396	479.90
	Rutin	8.9930	0.2558	4968	551.60
	Ursolic acid	10.2220	0.0321	4860	548.70
	Albendazole	6.1310	32.0565	3904	443.90
	DEC	5.7720	58.7581	3398	353.70

## Data Availability

The 2D electrophoresis gel images and proteomics data have been deposited to Zenodo, available at https://doi.org/10.5281/zenodo.13968369.

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
