# Peer review of "HRAMS Proteomics Insights on the Anti-Filarial Effect of Ocimum sanctum: Implications in Phytochemical-Based Drug-Targeting and Designing"

_proteomes, 2024, doi:10.3390/proteomes13010002_

Round 1

Reviewer 1 Report

Comments and Suggestions for Authors

Dear authors,

the manuscript at hand concerning your investigation of O. sanctum extract and its anti-filarial effects is well written and clear. While the topic is of intrest, there are some issues that should be addressed:

1) The preparation of the O. sanctum extract is not described in the methods section, just later in the manusript it is told that it is an alcoholic extract, but details are missing.

2) In line 197 it should read "(NH4)2SO4" rather than "NH4SO4".

3) In paragraph 2.12 is written that significance of regulation was judged by fold-change. However, significance is clearly a statistical value given by statistical tests such as ANOVA or t-test or similar.

4) Detailed infomration concerning the trypsin used should be given (vendor, activity) as the used amount seems to be rather high.

5) More details for the LC-gradient should be given.

6) The abbreviation "PC" for protein carbonyls should be changed as "PC" is the common abbreviation for phosphatidylcholines.

Best Regards!

Author Response

 1)  The preparation of the O. sanctum extract is not described in the methods section, just later in the manuscript it is told that it is an alcoholic extract, but details are missing.

Response:  Method of extract preparation of EOS is now described in materials and methods section. (Line no. 72-80).

 2)  In line 197 it should read "(NH4)2SO4" rather than "NH4SO4".

Response:  Ammonium sulphate formula has been corrected. (Line no. 174).

3)  In paragraph 2.12 is written that significance of regulation was judged by fold-change. However, significance is clearly a statistical value given by statistical tests such as ANOVA or t-test or similar.

Response:  Student’s t-test was used to calculate the statistical difference between the control and treated groups. The Table 2, with asterisks denoting the level of significance. A p-value of < 0.05 was considered significant, with the following notation: ***P < 0.001 indicating highly significant differences, **P < 0.01 showing moderate significance, and *P < 0.05 representing a statistically significant difference. (Line no. 307-308).

4) Detailed information concerning the trypsin used should be given (vendor, activity) as the used amount seems to be rather high.

        Response:  Correction has been done in materials and methods section. (Line no. 188-190).

 5) More details for the LC-gradient should be given.

        Response:  Now it has been added in materials and methods section. (Line no. 195-206).

 6) The abbreviation "PC" for protein carbonyls should be changed as "PC" is the common abbreviation for phosphatidylcholines.

Response:  Abbreviation of Protein carbonyl has been removed now. (Line no. 271-273).

Best Regards!

Thank You

Reviewer 2 Report

Comments and Suggestions for Authors

Summary

Mishra et al evaluated the anti-filarial potential of the ethanolic extract of Ocimum sanctum (EOS) against S. cervi that causes lymphatic filariasis (LF). The authors found that EOS significantly reduced parasite viability and induced oxidative stress. Proteomic analysis revealed altered expression of proteins involved in energy metabolism and stress response. Molecular docking indicated possible interactions between several EOS bioactive compounds and differentially expressed proteins (DEPs) of parasites when treated by EOS. These findings suggest that O. sanctum could be a plant-based option for developing new LF therapies.

The manuscript contains several unclear points and discrepancies. Please find the following issues regarding the manuscript bellow. I recommend a major revision if this manuscript is to be considered for publication in Proteomes.

Major comments:

1.     Could the author clarify the preparation and dilution process for the ethanolic extract of Ocimum sanctum (EOS) to achieve the specified concentrations (125, 250, and 375 μg/mL)? The authors mention in the Methods section that “worms incubated in maintenance medium containing DMSO (0.37%) acted as the vehicle control.” Does this indicate that DMSO was used as the solvent for EOS? If so, was the extract dried down before being resuspended in DMSO, and could this process introduce a bias by potentially losing volatile constituents? If drying down was not performed, how was the ethanol controlled in the experiment?

2.     On page 8, lines 322 to 325, the authors state: “Identified proteins spots that were increased were mostly stress-responsive proteins and chaperons like, Heat shock protein 70 (HSP 70), HSP 18, p27, antioxidant proteins Glutathione-S-transferase (GST), Thioredoxin domain containing protein 3 homolog protein.” However, in Table 2, p27 and Glutathione S-transferase show negative fold changes, and HSP 18 is missing. This discrepancy appears to be an error.

3.     Did the authors include a loading control for the gel analysis? Also, why does the ladder appear much denser in the control group?

4.     Regarding the biological processes of the 9 interrelated DEPs, what do the percentages following each GO term represent?

5.     The authors did not clarify the purpose or rationale for conducting molecular docking analyses, nor did they explain their selection of specific small molecules for docking when these molecules were first introduced. This makes the flow somewhat confusing. The authors identified potential interactions between several small molecules and key DEPs, but the significance of these interactions is unclear. Can these interactions be validated experimentally? Is it possible to perform functional assays to demonstrate potential pharmaceutical effects?

Minor comments:

1.     If the authors have the raw mass spectrometry files, depositing them into a data repository would enhance data transparency and benefit readers.

2.     On page 16, line 548, period “.” is missing after “in future”.

3.     Please make sure the panels in Figure 5 are clear and well aligned.

Author Response

Major comments:

  1. Could the author clarify the preparation and dilution process for the ethanolic extract of Ocimum sanctum(EOS) to achieve the specified concentrations (125, 250, and 375 μg/mL)? The authors mention in the Methods section that “worms incubated in maintenance medium containing DMSO (0.37%) acted as the vehicle control.” Does this indicate that DMSO was used as the solvent for EOS? If so, was the extract dried down before being resuspended in DMSO, and could this process introduce a bias by potentially losing volatile constituents? If drying down was not performed, how was the ethanol controlled in the experiment?

Response:  Method of extract preparation of EOS is described in materials and methods section. Dilution of EOS is also mentioned. (Line no. 72-80).

  1. On page 8, lines 322 to 325, the authors state: “Identified proteins spots that were increased were mostly stress-responsive proteins and chaperons like, Heat shock protein 70 (HSP 70), HSP 18, p27, antioxidant proteins Glutathione-S-transferase (GST), Thioredoxin domain containing protein 3 homolog protein.” However, in Table 2, p27 and Glutathione S-transferase show negative fold changes, and HSP 18 is missing. This discrepancy appears to be an error.

Response:  Now the correction has been done in text. (Line no.292-296).

  1. Did the authors include a loading control for the gel analysis? Also, why does the ladder appear much denser in the control group?

Response:  Equal amounts of protein were loaded between the control and treated gels prior to analysis, ensuring a fair comparison of protein levels between the two groups. The same amount of molecular weight ladder was also used in both the control and treated lanes. Therefore, any observed differences in protein bands should reflect genuine biological differences between the control and treated groups, rather than artifacts related to sample loading or ladder application. We maintained consistency in control and treated groups by following the same protocol for electrophoresis and staining. The experiment was repeated twice and the results are based on mean of both experiments. Raw data is deposited in zenodo.org (https://doi.org/10.5281/zenodo.13968369).

  1. Regarding the biological processes of the 9 interrelated DEPs, what do the percentages following each GO term represent?

Response:  GO term represent the Gene Ontology of DEPs participated in biological processes with percentage. Correction is done in the legend of figure 3.

  1.  The authors did not clarify the purpose or rationale for conducting molecular docking analyses, nor did they explain their selection of specific small molecules for docking when these molecules were first introduced. This makes the flow somewhat confusing. The authors identified potential interactions between several small molecules and key DEPs, but the significance of these interactions is unclear. Can these interactions be validated experimentally? Is it possible to perform functional assays to demonstrate potential pharmaceutical effects?

Response:  - The purpose for conducting molecular docking analysis is now added. (Line no.).

                      The selection is now explained. (Line no. 450-464).       

                      The significance of these interactions is discussed. (Line no.).

                      Yes the laboratory would be performing functional assays soon. However the assays would need much time for experiments and analysis, hence we could not be able to include that now.

Minor comments:

  1. If the authors have the raw mass spectrometry files, depositing them into a data repository would enhance data transparency and benefit readers.

Response:  Yes, it is deposited and data is available at https://zenodo.org/ with DOI

https://doi.org/10.5281/zenodo.13968369

  1. On page 16, line 548, period “.” is missing after “in future”.

Response:   Correction has been done. (Line no.512).

  1. Please make sure the panels in Figure 5 are clear and well aligned.

Response: Correction has been done.

Round 2

Reviewer 2 Report

Comments and Suggestions for Authors

The authors addressed my concerns in the revision. Recommend acceptance.